# Spatially Structured Optical Pump for Laser Generation Tuning

**DOI:** 10.3390/nano14010049

**Published:** 2023-12-23

**Authors:** Gabrielius Kontenis, Darius Gailevicius, Victor Taranenko, Kestutis Staliunas

**Affiliations:** 1Laser Research Center, Vilnius University, Saulėtekio Ave. 10, LT-10223 Vilnius, Lithuania; gabrielius.kontenis@ff.vu.lt (G.K.); kestutis.staliunas@icrea.cat (K.S.); 2Branch of Applied Optics at the Institute of Physics, National Academy of Sciences of Ukraine, 10G, Kudriavska Str., 04053 Kyiv, Ukraine; 3Department of Physics, Universitat Politècnica de Catalunya, Rambla Sant Nebridi 22, 08222 Terrassa, Spain; 4ICREA—Institució Catalana de Recerca i Estudis Avançats, Passeig Lluís Companys 23, 08010 Barcelona, Spain

**Keywords:** spatial beam shaping, modulated pump, MECSEL

## Abstract

The goal and essential parameter of laser light conversion is achieving emitted radiation of higher brightness. For many applications, the laser beam must have the highest available beam quality and highest achievable power. However, lasers with higher average power values usually have poorer beam quality, limiting the achievable brightness. Here, we present a method for improving the beam quality by using a spatially structured optical pump for a membrane external cavity laser resonator. An increase in brightness is achieved under fixed focusing conditions just by changing the pump intensity profile. A controllable output laser mode can be achieved by using a dynamically changing pump pattern.

## 1. Introduction

Since the first construction of lasers, many improvements have been performed to meet the growing number of applications. Generating radiation with different wavelengths, controlling and decreasing the pulse duration, increasing the conversion efficiency, and narrowing the generation spectra are just some of the directions of laser development. One property of the laser that is steadily improving is the high spatial quality $M^{2}$ of the beam [1]. It characterizes how a particular beam parameter product (proportional half-angle divergence θ times the smallest observable radius ω/2) compares to that of a diffraction-limited Gaussian beam. In addition, the laser beam output power *P* divided by the beam quality factor $M^{2}$ defines the beam brightness Br. Assuming a circular Gaussian beam characterized in two directions, *x* and *y*, $Br\propto P/M_{x}^{2}M_{y}^{2}$ [2,3]. In the context of classical laser concepts, an increase in the output power might occur together with a decrease in the beam quality (an increase of $M^{2}$), and therefore, the brightness does not scale trivially. Typically, it decreases with an increase in the pump as observed for microchip solid-state lasers [4,5] and semiconductor systems [6,7]. This is because it exceeds the generation threshold for higher-transverse modes characterized by higher divergence [8,9].

The beam quality is important in many applications, as shown by micromachining [10,11] and others [12,13]. There are, however, other contexts where beam spatial quality is not essential [14,15]. There are cases where a Gaussian shape is not the best option for applications, and other types of transverse intensity patterns are desirable. For a more uniform laser ablation process as compared to a Gaussian pattern, a flat-top beam gives better results and more control [16,17]. For high-aspect-ratio structures, a Bessel beam is required [18]. However, classical resonators are designed to output a fundamental Gaussian mode and various beam conversion techniques are being used to convert the beam into the desirable pattern [19]. The aforementioned approach of transforming the freely propagating beam outside the laser [20,21] is not the only option. A more sophisticated option is to shape the pump beam to impose a different gain distribution and/or guide in the resonator [22]. Technically, this approach facilitates the interaction of light with the gain media without mechanical or electrical manipulation of the resonator. This type of (“extracavity”) modulation is more suitable for the better control of thermal effects [23] and energy extraction [24]. In addition, it is possible to shape the beam inside the cavity (“intracavity”), forcing certain modes to resonate and dominate over others. Due to the filtering nature of the resonator [25,26,27,28], this method of modulation is mostly used for mode selection where gain or loss can be spatially controlled to select the desired mode [29,30,31,32,33,34] of the laser output and obtain a higher energy extraction efficiency from the lasing medium [35,36] by using top-hat or flat-top beam profiles when the beam propagates through the gain region. Another benefit of intracavity modulation is the ability to form flat-top beams that are invariant under propagation. This can be achieved by the mixing of $LG_{00}$ and $LG_{01}$ beams [37], whereas reshaping the propagating beam using diffractive or refractive methods results in a flat-top profile with a uniform intensity pattern only in the vicinity of the focusing lens focal plane. Amplitude modulation inside the laser cavity has been shown by inducing losses with a wire to produce Hermite–Gaussian modes [38].

The flexibility of the surface-emitting semiconductor laser architecture provides a unique opportunity to test the aforementioned beam shaping/brightness improvement techniques. The compact vertical cavity surface-emitting lasers (VCSELs) are gaining more and more interest due to their potential applications in data transfer and metrology [39]. Their ability to emit light from the surface instead of from the edge (in more common semiconductor lasers) allows them to be clustered into large 2D arrays. A VCSEL comprises a thin amplifying medium made from quantum wells with Bragg mirrors formed on both sides to form a resonator [40,41]. In these lasers, transverse modes form due to the pump profile and charge distribution in the cavity [42]. The large-aperture VCSEL as well as the 2D arrays although capable of high power emission provide multimode beams and thus low brightness. However, optically pumped vertical external cavity surface-emitting lasers (VECSELs) [43] and their newer versions, the membrane external cavity surface-emitting lasers (MECSELs) have the ability to have both a higher beam quality and higher output powers [11,44,45] when compared to VCSEL monolithic designs. These versions are typically pumped with a laser diode featuring its specific transverse intensity profile. Additionally, at least one of the cavity mirrors is separated from the structure, and a concave output coupling mirror is used to stabilize the cavity and shape the mode. These structures should have better power scalability due to their spatially uniform optical pumping, much lower thermal stresses [46], and better thermal management [47].

Photonic crystal (PhC) spatial filters are a potential solution to suppress multimode operations in microcavity semiconductor surface-emitting lasers, increasing the output beam spatial quality and its brightness [48,49,50,51]. The other alternative is photonic crystal structures integrated with the gain medium, as in photonic crystal surface-emitting lasers (PCSELs) [52,53,54,55,56]. PCSELs are well-known for their ability to generate a high power output with an impressive beam quality, particularly in single-mode operation, which is a major improvement compared to conventional surface-emitting lasers. However, due to the complex photonic crystal structures and demanding fabrication techniques, they have difficulty sustaining this high power output.

Lastly, the emission brightness can be improved through modification of the pump beam. The periodic gain structures can be created by interfering multiple pump beams on the gain media at different angles. The dimensionality of the photonic structure can be controlled by changing the polarization of the interfering light, allowing dynamic control of the structure [57,58]. Although multiple solutions that affect the transverse mode structure exist, they usually do not include the manipulation of the pump profile. To address this, here, we demonstrate that the brightness of a semiconductor membrane external cavity surface-emitting laser mode of operation can be improved via structured optical pumping. The change in brightness is an indirect observation of transverse mode interaction with the periodic gain structures intentionally induced in the cavity. For this purpose, we use a modified diode-pumping beam approach and a surface-emitting membrane semiconductor active medium to characterize our particular approach.

## 2. Methodology and the Laser Setup

### 2.1. Models for Phase Mask Selection

We used amplitude modulation to shape the intensity profile of the pump beam. By shifting the phase of close-by pump beam regions, destructive interference forms beam shapes with a dip in intensity. A pattern with a periodically changing intensity acts as an array of coherent laser sources similar to [59]. With this approach, the efficient, coherent combination of 16 solid-state laser channels operating in single transverse Gaussian modes has been shown to have an efficiency of 87%. The reported beam quality of the combined beam was $M^{2}=1.3$, meaning that the beam quality remained the same but with an increase in power. Talbot resonators [60,61] have been shown to phase-lock separate yet coherent sources. Non-phase-locked lasers usually generate high power outputs but are not very bright, whereas phase-locked lasers also offer a good beam quality. The Talbot effect is a phenomenon in which a coherent periodic structure replicates itself at periodic intervals. In the ideal case, the initial field would reimage itself after the Talbot distance:(1)$$Z_{T}=\frac{2d^{2}}{\lambda },$$
where λ is the wavelength and *d* is the transverse spatial period. This effect has a major influence on the structure of the transverse mode when it arises inside the cavity of a laser array [62]. In our case, the pump beam’s linear periodic intensity distribution acts as multiple coherent emitters. The spacing period can be adjusted by altering the relevant hologram. We take a similar approach to that employed in [63], where two-dimensional, regularly arranged electrical contacts were produced on a broad-area semiconductor (BAS) amplifier. In the case of optical pumping, the spatial structure of the pump beam determines the transverse periods $\Lambda_{⊥}$, and the length of the resonator is used as the longitudinal period $\Lambda_{\|}$. As the laser oscillates in the cavity with each round trip, the beam is affected by the intensity distribution of the pump source. The pump source determines the gain distribution and, therefore, the amplitude modulation. This should result in a decrease in divergence due to the feedback between the pump structure and the resonating mode, which can be represented by the parameter *Q*:(2)$$Q=\frac{2\Lambda_{⊥}^{2}n}{\lambda \Lambda_{\|}}.$$

The concept of a periodically structured pump integrated into a laser cavity is based on selective deflection of the angular components of light passing through a two-dimensional photonic structure. The angular components that are resonant with the transverse and longitudinal periodicities of the structure are diffracted efficiently and deflected away from the zero-diffraction order of the transmitted beam. Previous studies have demonstrated that periodic gain/loss modulation on the wavelength scale can lead to specific beam propagation effects, such as self-collimation and spatial (angular) filtering. For longer resonators, the beam profile is mainly determined by the most amplified mode, while for shorter or moderate lengths, a comprehensive analysis of the mode growth reveals that other modes also contribute to the final beam shape. By selecting the appropriate spatial periods, it is possible to narrow the central far-field component while significantly improving the spatial structure of the amplified beam. Additionally, by shaping the pump beam, a good spatial overlap between the pump and the desired mode can be achieved, thus forcing the laser oscillations in that transverse mode.

### 2.2. Experimental Setup

We used a 400 µm core diameter fiber-coupled λ=808 nm laser diode as the pump source. It could be run under continuous-wave (CW) or quasi-continuous-wave (QCW) regimes using pulse width modulation. A thin membrane external cavity surface-emitting laser (MECSEL) with a thickness of 1 µm was used as the active medium, as shown in Figure 1b. We used liquid crystal on a silicon spatial light modulator (SLM) to act as the beam-shaping element of the pump beam. It was a two-dimensional 1980×1024 pixel array with a pixel size of 8 µm that allowed the control of the phase of the laser beam by delaying it locally with each pixel. This made it possible to distort the laser wavefront, which is represented by a surface characterized by the same phase. This changed the further propagation of the laser beam, and due to the interference, a dynamically controlled and variable intensity distribution formed. It is possible to locally reduce the intensity and perform amplitude modulation of the laser with the SLM. This can be performed by exploiting the fact that adjacent pixels shift anti-phase, allowing destructive interference generation. A magnifying telescope with two lenses, 1000 mm and 30 mm/15 mm, was used in a 4F setup to scale and relay the structured intensity distribution to the active area of the MECSEL chip. A QCW regime was used for experiments with a pulse duration of 200 µs and a repetition rate of 500 Hz. When a flat output coupling mirror is used, the entire resonator is only stabilized by the thermal lens inside this very thin MECSEL chip. Although the structure is 1 mm thick with an active area of just around 1 µm and fast heat dissipation, the thermal lens still arises and stabilizes laser operation. A flat output coupling mirror (OC) with a reflectivity of 99% was placed with an air gap of 200 µm to give a total physical resonator length of 1.2 mm.

The telescopes used in the experiment were composed of magnified 4F systems, as shown in Figure 2a. The corresponding pixel count in the amplitude mask on the SLM screen corresponded to a certain size in µm in the focal plane and scaled linearly. The various holograms used in the experiment are shown in Figure 2b. The main types comprise ring holograms that selectively attenuate the edges of the pump beam or periodic line patterns with varying widths that etch out linear intensity regions.

## 3. Results

To see the effects of the structured pump on the active layer itself, the spontaneous emission of the beam shape was visualized onto a CCD beam profiler by using an imaging lens in a double focal length setup (2F2F). Figure 3 shows that the pump beam shape is visible on the active media. However, the contrast between intense and dark regions is not as clear as on the pump beam. This can be due to the diffusion of the carriers in the active areas of quantum wells. This limits the pump structures’ density before diffusion equalizes the intensity distribution.

Due to a large selection of possible pumping intensity shapes, some basic intensity distributions were chosen (Figure 2). One way of modifying the pump beam is by removing the central intense region—a “Disk” hologram achieves this. Alternatively, a “Ring” hologram can apodize and remove the edges. Additionally, some periodic structures can be formed by a repeating linear or 2D varying modulated and non-modulated regions. Because we use amplitude modulation, the part that sees destructive interference also diffracts out of the optical chain’s optical axis and does not propagate towards the active media. Therefore, if we normalize each used beam shape with the actual optical power impinging on the MECSEL chip, we see that the conversion efficiency remains unchanged regardless of the shape (Figure 4). The beam size on the MECSEL chip was 175 µm, as determined with a 66.6× reducing telescope made from a pair of lenses with 1000 mm and 15 mm focal lengths. The acquired conversion efficiency was η=25%. All parameters were measured in the same manner, so systemic errors were identical. The beam width of the output beam was measured at the focal plane of a 50 mm focal lens. The lens was placed 150 mm from the resonator’s OC in all cases. The far field intensity distribution was measured at two planes, 5 mm and 10 mm from the focal spot.

The oscillating resonator’s fundamental mode must be much smaller than the pump beam in order to allow higher-order beam shapes to oscillate. If the fundamental oscillating modes and pump beam sizes match, no higher modes are formed and just a decrease in power is present due to the lower coupling of the beam. They all follow the same pump power generation path with similar divergence and focusing parameters. The output beam mostly remains in $TEM_{00}$ mode with a slight decrease in power in the central region with higher pumping outputs (Figure 5). How the far-field intensity distribution changes with an increase in the pump power is shown in (Appendix A). The shape oscillates by becoming more elliptical in one of the axes, then becoming circular, and then elliptical in the perpendicular direction. At higher powers, the maxima of the output beam flattens out. It is no longer a pure $TEM_{00}$ distribution; there are higher modes mixed in. This gives rise to a decrease in the overall brightness.

For the brightness calculations of the laser beam, the expression used was
(3)$$Br=\frac{P}{A\Omega }=\frac{P}{\pi \theta_{x}\theta_{y}\omega_{0x}\omega_{0y}},$$
where $\theta_{x}$, $\theta_{x}$ are the beam’s divergence and $\omega_{0x}$$\omega_{0y}$ are the beam’s waist diameter at the focus in the x and y directions, respectively. From Figure 6, we can see a brightness peak at a pump power of around 350 mW. Around this power level, the far field of the generated beam starts to have a flatter intensity distribution. A further increase in power starts to form an intensity dip in the center. The dip could indicate the formation of higher-order modes. Because MECSELs have heat spreaders on both sides of the active layer, they inherently form a microcavity inside the structure. This has the effect of forming multiple emission peaks at different wavelengths. Therefore, identifying multimode operations from spectral data is not easy. From power conversion characteristics (Figure 4), a linear relation between the output and pump power is visible. Saturation was not reached in this experiment. The only source of a decrease in brightness was the degradation of the quality of the beams. The divergence of the beams increased almost linearly with the oscillations due to the shifting ellipticity in the far field, while the beam’s waist diameter (Figure 7) at the focus narrowed. As higher modes became excited, the spot size started to increase.

The 1D periodic line pump divergence in Figure 8 shows a peculiar phenomenon in which the beam divergence does not appear to change at the period of 12 µm. Suppose we look back at Equation (Equation 2), the cavity comprises a 1 mm long MECSEL structure with a 0.2 mm air gap separating the semiconductor chip and the outcoupling mirror. The detailed structure of the MECSEL is unknown (as far as we can tell, it is proprietary), but assuming an average refractive index of 3, we show that, for the case where $\Lambda_{⊥}=12$ µm, the longitudinal period is $\Lambda_{\|}=812$ µm, and for the case where $\Lambda_{⊥}=15$ µm, $\Lambda_{\|}=1269\;$µm. Therefore, the stabilization of divergence could be attributed to the matching of the longitudinal and transverse periods. Here, using 6 µm and 12 µm periods is too far from the Q=1 relation. However, the power scaling did not generate clearly visible $LG_{01}$ and higher-order modes.

**Curved resonator setup.** To see a change in the output beam shape, a curved OC with a radius of curvature of 100 mm was used instead of the flat mirror mentioned above. This resulted in a flat–concave resonator setup. The resonator length of L = 95 mm gave a mode size on the MECSEL chip of 2ω=168 µm. The pump spot diameter resulting from a 30 mm lens was 340 µm, larger than the curved mirror case’s fundamental oscillating mode size. A curved mirror selects the modes that will oscillate inside a resonator, but using an elongated pump allows only modes above the threshold to lase. This shape is achieved by a cylindrical lens phase hologram formed on an SLM. Then, we can select sets of transverse modes by rotating the pump shape with the holograms. The resonator itself determines the shape. An asymmetric pump profile in an ellipse gives rise to asymmetrical modes, essentially getting an elliptical output that follows the pump pattern (Figure 9). The exact output shape does not rotate with the pump precisely; there is some asymmetry in the resonator itself, and one orientation gives rise to a larger ellipticity than the other.

Another tested beam shape was in the form of transverse LG modes with different topological charges. Such modes were formed using phase holograms with azimuthally changing phase offset, where the topological charge is given by the number of times the phase offset reaches the 2π value. While the pump beam was an ever-higher-order LG mode with an increasing topological charge (Figure 10), the generated intensity pattern began to form four maxima in the shape of a spatial mode $TEM_{11}$ instead of a larger ring. The polarization of the output beam remained linear and did not split between the modes. A slight intensity mismatch between the maxima in Figure 10i,j could have been due to asymmetries in the optical scheme.

## 4. Materials and Methods

The pump laser used for the experiment was a K808DAERN-30.00W electrically pumped diode laser operating at 808 nm from BWT BEIJING and coupled witha 400 µm core multimode fiber. The active media used for the setup was a “21 semiconductors” supplied 21S-M1064-496 MEXL chip that uses an 808 nm pump and outputs 1064 nm. The chip was soldered with an HR Bragg mirror side to the brass plate. A HoloEye SLM PLUTO-2.1-NIR-113 was used for beam shaping with a two-dimensional 1980×1024 pixel array with a pixel size of 8 µm. A Peltier element with a copper plate was used to mount and cool the MECSEL chip during operation. A CCD beam profiler (WinCamD-UCD15, DataRay Inc., Redding, CA, USA) was used to measure the beam. An inline pumping scheme was used with the pump beam shape modulated by the SLM using predominantly checkerboard patterns for amplitude modulation.

## 5. Discussion and Outlook

While being a dynamic optical element, the SLM does give the ability to tune and change the output characteristics of the laser in real time. Using an SLM for the structured pump has drawbacks, because the pixel size is much larger than the pump wavelength, and linearly polarized light must be used as the input. High amounts of magnification are needed to use the entire SLM screen for beam shaping, and focusing it down to appropriate power densities for laser pumping requires a high ratio of focal lengths for the 4F system, making the entire system bulky. This method works as a proof-of-concept with the ability to tune and dynamically modulate the output but not as the final design of a compact semiconductor laser. The pixel size can be somewhat remedied by using SLMs with smaller pixels. Using smaller pixels in liquid crystal SLMs can cause an increase in crosstalk between them. This is because the pixels form a continuous grid, and the voltage applied to neighboring pixels affects the phase shift of a single pixel. This information is detailed in [64]. In this experiment, amplitude modulation was predominantly used; other devices, such as a digital micromirror device (DMD), might be a better and faster solution.

The dominant mechanism for the formation of the dominant mode is defined by the resonator structure/configuration and not the pump beam structure, but resonator modes that do not overlap with the pump beam cannot resonate. A change in the pump beam can selectively choose all modes that overlap with the pump beam. Because the resonator’s structure still determines the final oscillating mode, carrier diffusion and resonator feedback smooth out any pixel crosstalk and intensity fluctuations. Therefore, smaller feature sizes in the pump beam lead to small modulation depths and do not influence the laser output.

The amplitude modulation approach for textured pumping does induce a loss in the pump’s optical chain, and the achieved spatial structure of the pump beam does not diminish the output power conversion efficiency. The output beam comes with a slightly reduced divergence and tighter focusing. After the beam’s brightness has peaked, textured-pump-induced generation features a less rapid reduction in brightness. The observed increase in brightness is obtained because the ring-shaped holograms limit the pump beam diameter on the active medium (such as closing a physical aperture inside the resonator). The total intensity of the beam decreases weakly from the edges; however, the beam divergence can be significantly reduced. If the aperture is too small, it will affect the lowest transverse modes, decreasing the power. The limiting factor in this experiment was the available diode pump power. Therefore, a pure single higher-order mode operation could not be achieved.

Using a lossless approach to create the required pump intensity pattern could further increase the efficiency and usefulness of this method. Further experiments and theoretical simulations are needed to understand better the complex interaction of the non-trivial pump shape and resonant mode coupling.

## 6. Conclusions

In conclusion, we have shown the output mode modification for two cases: flat–flat and flat–curved resonator configurations. We evaluated the generated beam’s output power and propagation parameters. We demonstrated this principle on a diode-pumped membrane external cavity surface-emitting laser, where the pump beam’s intensity profile was shaped. The flat–parallel configuration of the resonator and pumping from the end favored a Gaussian mode oscillation. We generated a slightly reduced divergence and sharper focusing, indicating an improvement in the beam quality for a given MECSEL output power when the pump was concentrated in a smaller area. Therefore, the resonant mode size decreased in the flat–flat resonator configuration. We propose a feasible and dynamic scheme to tailor and control the complex spatial dynamics of MECSELs. The mechanism for divergence stabilization has not been fully investigated. A possibility could be that the intensity dips introduced in the pump profile can act as attractors, concentrating the resonating beam’s power onto the optical axis and coupling better with the fundamental mode.

## Figures and Tables

**Figure 1 nanomaterials-14-00049-f001:**
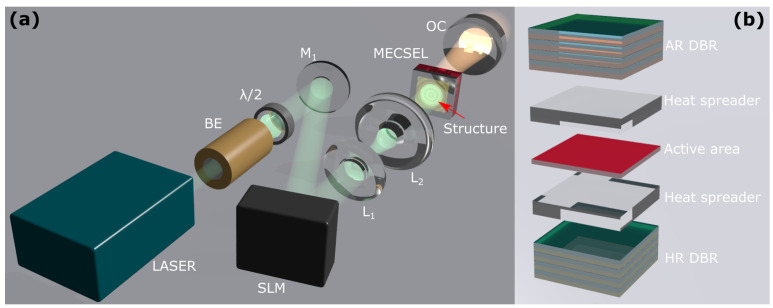
(**a**) The setup used in this work. Annotations: λ/2—half wave phase plate, M—mirror, BE—beam expander, SLM—spatial light modulator, L—lens, MECSEL—membrane external cavity surface-emitting laser chip, OC—output coupling mirror. (**b**) Displays the simplified structure of the MECSEL chip. An active area is encased in heat spreaders and distributed Bragg reflectors (DBRs), where one side acts as the resonator mirror (HR) and the other reduces the resonator losses by reducing reflections (AR).

**Figure 2 nanomaterials-14-00049-f002:**
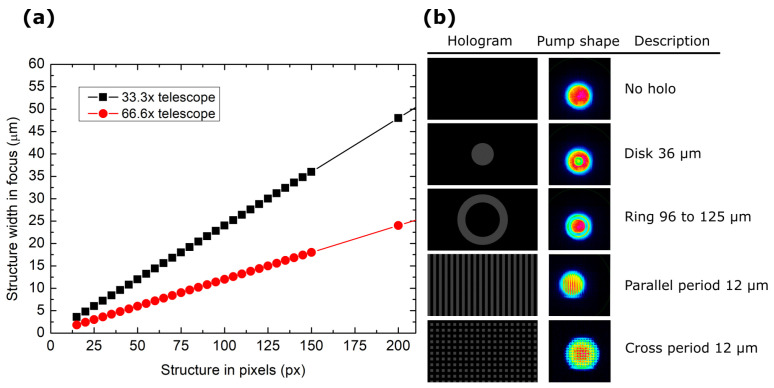
(**a**) The number of pixels used to generate an amplitude modulation in the near field corresponding to a certain width in the magnified region. (**b**) The holograms with their corresponding code names and the resultant pump intensity profile.

**Figure 3 nanomaterials-14-00049-f003:**
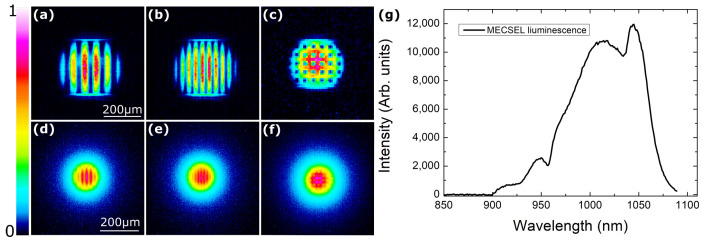
(**a**–**c**) The intensity patterns of the pump beam and (**d**–**f**) the corresponding spontaneous emission pattern on the MECSEL active layer. The scaling used is 33.3×. (**g**) The typical spontaneous emission spectrum, as observed at a pump power of 200 mW.

**Figure 4 nanomaterials-14-00049-f004:**
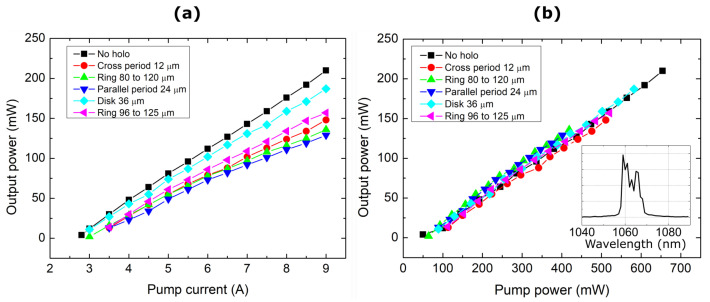
(**a**) The output power based on the pump diode current; (**b**) the MECSEL laser output power versus the pump beam power incident onto the chip. In the inset on the lower right is a snapshot of the typically observed spectrum of the output beam.

**Figure 5 nanomaterials-14-00049-f005:**
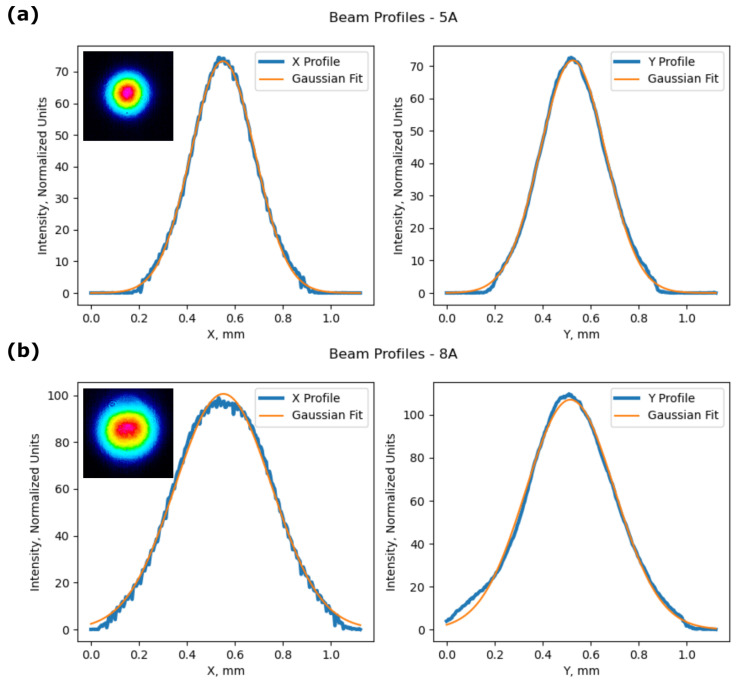
The intensity distribution of the generated beam in the X and Y cross-sections with an average pump power on the active chip area of (**a**) 5A—291 mW and (**b**) 8A—586 mW.

**Figure 6 nanomaterials-14-00049-f006:**
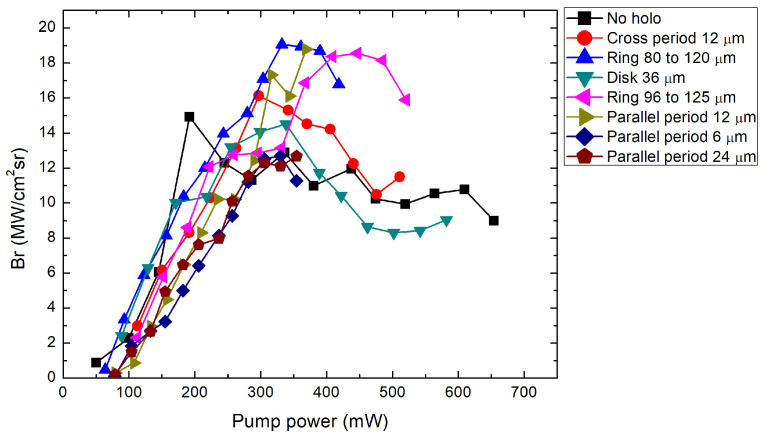
The MECSEL laser’s brightness dependence on the pump beam average power in the X and Y directions respectively. “No holo” is the case where the SLM was used as a reflecting mirror with no modulation.

**Figure 7 nanomaterials-14-00049-f007:**
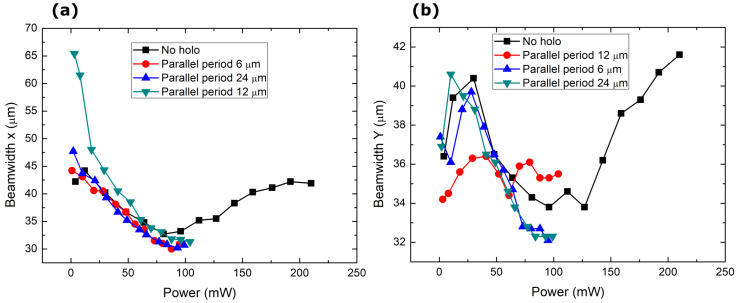
Change in the focal beam size of the generated laser radiation with power (**a**) in the X direction, and (**b**) in the Y direction.

**Figure 8 nanomaterials-14-00049-f008:**
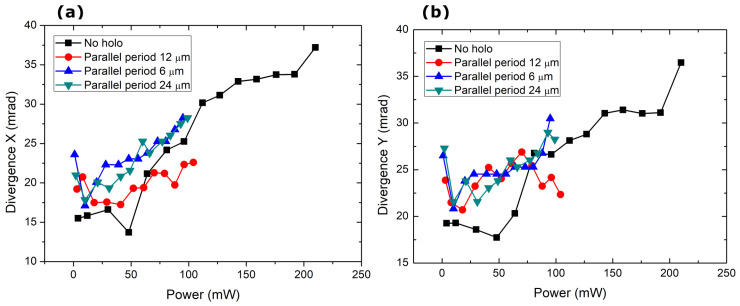
Change in the divergence of the generated laser radiation with power in the (**a**) X and (**b**) Y directions.

**Figure 9 nanomaterials-14-00049-f009:**
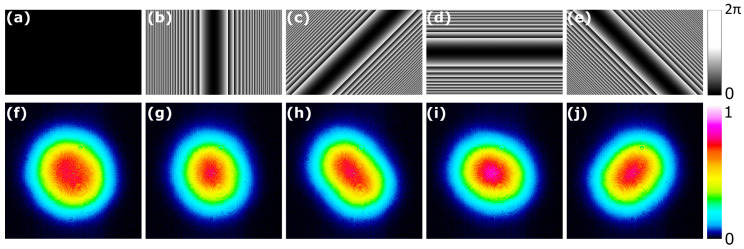
(**a**–**e**) The holograms used to shape the diode pump beam; (**f**–**j**) the laser output beam’s far-field shapes as a result of a rotation in a cylindrical hologram on the pump beam. The corresponding rotations are 0, 45, 90, and 135 degrees.

**Figure 10 nanomaterials-14-00049-f010:**
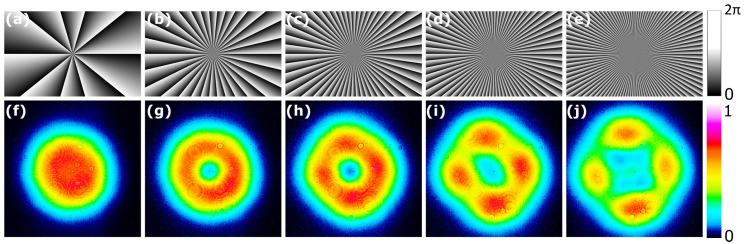
The holograms used to shape the pump and the resulting MECSEL lasing beam output in the far field. (**a**–**e**) The pump beams’ phase hologram; (**f**–**j**) the corresponding generated distribution. The corresponding topological charges are l=10, 30, 50, 70, 100.

## Data Availability

Data are contained within the article.

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
