# Peer review of "Spatially Structured Optical Pump for Laser Generation Tuning"

_nanomaterials, 2023, doi:10.3390/nano14010049_

Round 1
Reviewer 1 Report
Comments and Suggestions for Authors
The article “Spatially structured optical pump for laser generation tuning” written by Gabrielius Kontenis with co-authors can be published in Nanomaterials after appropriate minor revision. A sequential complete analysis of the output mode modification in the case of flat-flat one and flat-curved resonator configurations – it was evaluated the generated beam’s output power and propagation parameters.
The presented results are of scientific interest however I leave it to the journal editor's decision, but in my opinion the topic of the article is not within the main topics of the journal. Also the work is more technological in nature than fundamental.
Authors should pay attention to the following remarks that need to be answered and make the relevant changes to the article.
1. It is necessary to specify and describe in more detail the specifics of phase masks (holograms) formation shown in Figures 9 a-e, 10 a-e. Provide a detailed description of the SLM characteristics and how it is controlled. Is it a commercially available SLM or of your homemade SLM? Were holograms simulated to obtain a specific modulation of laser beams? What was the motivation for utilising a particular type of hologram to achieve ring-type and elongated beams?
2. It should be clarified that with lasers advent there is active research not only on power increasing and higher beam quality but also on the development of ultra-narrow bandwidth lasers, wavelength design and tuning, control of pulse duration, operation efficiency, etc.
3. The third remark is of a recommendatory nature. In case the authors consider it appropriate to mention such a demanded problem in the field of nanophotonics, quantum computing and communication as light control by light. In the work context, it is reasonable for me to consider and compare the results and peculiarities of beam acts as multiple coherent emitters formation in the case of self-diffraction processes described in previously published studies, e.g. on dynamic photonic crystals (https://doi.org/10.1364/OL.389127, https://doi.org/10.1134/S0021364016220136) or on circular aperture (https://doi.org/10.1016/j.physe.2012.05.028)

Reviewer 2 Report
Comments and Suggestions for Authors
Review of the paper „Spatially structured optical pump for laser generation tuning”
Paper describes the emission properties of MECSEL laser. Reports an excellent experiments which are well performed and explained. The conclusion are well formulated.
Weak points:
Described research has nothing to do with nanomaterials, especially with 2D or Carbon Nanomaterials. It is out of scope of the journal. Paper is written with poor English. Has to be corrected.
The text should be prepared with greater care. Naw the text is often difficult to understand. The colloquial language is used. English native speaker has to read the paper and correct the grammar or use commercially provided help by https://www.proof-reading-service.com/en/ , for instance.
In introduction I would suggest to provide information that your research is on “optical pumping”, “disc laser”, “diode pumping “ Take care to call particular laser always the same
VECSEL are not vertical surface emitting lasers
MECSEL are not membrane external cavity laser
What is “Surface-emitting lasers” L45
Cavity or resonator ?
Same errors and suggestions noticed by reviewer (not all present in the paper).
L13 “For high-power approaches large pump areas” may by “ For high-power application large pump areas…” would be better.
L16 “In order to further improve both laser parameters some aspects of the laser have to be modified.” Please remove sentence since it do not provide any information.
L17/21 “Also, a Gaussian shape isn’t the best option for all applications and other shapes are desirable. For a more uniform laser microfabrication, a flat-top beam gives better results and more control [2,3]. For high aspect ratio structures a Bessel beam is required [4]. However,
most of the time resonators are designed to output a fundamental Gaussian mode.”
Try to provide this information in new order “Most often the resonators are designed to output a fundamental Gaussian mode. However, a Gaussian shape isn’t the best option for all applications and other shapes are desirable. For a more uniform laser microfabrication, a flat-top beam gives better results and more control [2,3]. For high aspect ratio structures a Bessel beam is required [4].” For reader it is easier to understand if it first get general information then the exceptions.
If you write the information in other way the reader get filling that you deny yourself, that those exceptions are not important. This also suggest that your research is not important because you show results for a certain particular resonator.
L24 are you sure the words “(extracavity modulation)” should be in this line ?
L38 “ The newer microchip lasers..” newer in regards to what ?
L48 “ have higher beam quality and higher output power” in comparison to what ?
L47-L48 SSL disc laser are made from different materials than the semiconductor based MECSELs thus the information provided here should be more precise
L50 what is BA device
L55 compare the ref. [28] with https://theses.hal.science/tel-02049651/file/50665_SEGHILANI_2015_archivage_cor.pdf
and with papers of Montpellier team.
I would advise to concentrate the paper on the semiconductor based optically pumped MECSELs and VECSELs . Please get rid of the references to gas lasers and SSL, this will make introduction to your paper more clear. In conclusion at the end of the paper you can write that your reserch is universal for all disc laser.
L133 “To see the effects of the structured pump on the active layer itself, the luminescence of the beam shape was visualized onto a CCD beam profiler” The sentence is not precise enough. you have not see luminescence of the beam but spontaneous emission generated in an active region (QW). Would by good if you provide information what was the wavelength of the luminescence ?
L137 what is quantum well why didn’t you explain this when you describe the MECSEL L107 ?
L138 “This does give a limit to how dense the pump structures…” I do not understand what you mean “dense” here. May be “This does give a limit for spatial modulation of the pump beam profile “ would be better ?
L140 I do not understand the sentence “Due to the big parameter space various pumping intensity shapes were chosen for evaluation Figure 2.”
L141 “the part that see” should not be used in scientific paper, this is colloquial language which can be used when you talk with your colleagues,. But not written in scientific paper.
L142 “doesn’t” should not be used in scientific paper.
L145 “@ 808 nm wavelength” should not be used in scientific paper.
L159 “For a .gif” should not be used in scientific paper.
L166 Sentence “Around this power level, the far field of the generated beam started to have a flatter intensity distribution with a further increase in power giving an intensity dip in the center.” Is difficult to understand. Divide it to two. “…………….intensity distribution. The further increase in power gives…”
L168/169 “This could be the formation of higher-order modes” please show the laser spectra. High order modes will be easily visible in the laser spectra
L169 “From the power curves…” change for “From power conversion characteristics (Fig. 4)…”
L186 “has been talked about up until now.” should not be used in scientific paper, use “above-mentioned„
L231 Another sentences which is difficult to understand. Please add what is in bold “The dominant mechanism for the resonant mode formation is defined by the resonator structure/configuration and but not the pump beam structure, but However, resonator modes that don’t overlap with the pump beam can’t resonate. A change in the pump beam can selectively choose all modes that overlap with the pump beam.”
L236 “…become not a problem…” should not be used in scientific paper because reader do not know what “problems”. Be more precise.
There are many more to correct, but do not give up. Correct and send again to high impact factor journal dedicated to optics !
Comments on the Quality of English LanguageThe text should be prepared with greater care. Naw the text is often difficult to understand. The colloquial language is used. English native speaker has to read the paper and correct the grammar or use commercially provided help by https://www.proof-reading-service.com/en/ , for instance.
Round 2
Reviewer 2 Report
Comments and Suggestions for Authors
You have nice results describing the change in the laser brightness when a structured light is used for pumping a disc laser. The experimental data for sure should be presented in good scientific journal dedicated to quantum optics or photonics.
However there are weak points of your paper:
1. The introduction is the weakest part of the pater. The information you provides is a mess. It is not clear what ia general feature of VECSEL for instance and what ia a specific feature observed in particular experiments. It demonstrates authors small knowledge on the laser architecture and the physical mechanisms behind the beam quality. There is lack of the precision require in science. I would advise to concentrate your introduction on the optically pumped disc lasers or even microchips laser. The broad approach you took makes the introduction miss leading the readers. What you write is often not precise or simply untrue. This has to be changed. In your introduction it should be clearly stated emission/beam/ brightness of what kind of laser can be improved following your ideas.
2. There is missing the definition of brightness in introduction. In whole paper you should use a single measure of the improvements you deserve: beam quality, or brightness or high order modes discrimination.
3. You should not introduce your own names/definitions (L44). Surface emitting semiconductor lasers what it is ? Semiconductor surface emitting laser can be VCSEL, VECSEL, MECSELs, PhCSEL. Ref 26 paper by Guina et al., well describes what is the VECSEL, It is clear that the arrays can not be fabricated with VECSELs, ref. 28 by Broda et al., describes the MECSEL. VECSEL and MECSEL are not used for data transfer (L45) ref 22 by Alford et al. is not suitable here, do not support VECSEL application for data transfer.
4. The VECSEL and MECSEL use external cavity in most cases enclosed by concave dielectric mirror which position determine the beam quality. The dielectric mirror curvature sets the focus size. Focus size together with pump spot determine those laser emission power.
5. VCSEL and PhCSEL are monolithic. The VECSEL and MECSEL are opticaly pumped wheras the VCSEL and PhCSEL are electrically pumped. In case of the VCSEL the beam quality is limited by current crowding along the circular contact. This current crowding makes the spatial (lateral) gain distribution in VCSEL ununiform. The photonic crystal in PhCSEL is a way to improve the beam quality (but the mode structure of PhCSEL is more complicated issue). Ref 30 by Peckus et al. is no suitable here tray rather a paper by Susumu Noda leader of first PhCSEL research team Nature Photonics volume 8, pages 406–411 (2014), for instance.
6. VCSEL has two DBR from both side of active region. VECSEL has one DBR, MECSEL has no single one. MECSEL was called DBR-free laser by Yang, Zhou, et al. "Optically pumped DBR-free semiconductor disk lasers." Optics express 23.26 (2015): 33164-33169. https://opg.optica.org/oe/fulltext.cfm?uri=oe-23-26-33164
7. MECSEL name was introduced by Kahle what was accepted by the research community because of the symmetry with the previous designs VCSEL, VECSEL, MECSEL
Kahle, Hermann, et al. "Semiconductor membrane external-cavity surface-emitting laser (MECSEL)." Optica 3.12 (2016): 1506-1512. https://opg.optica.org/optica/fulltext.cfm?uri=optica-3-12-1506
8. PhCSEL state for Photonic Crystal Surface Emitting Laser
9. I am nor sure whether the laser you describe should be called MECSEL. You use a semiconductor membrane but the cavity you build is different then standard used by Yang, Kahle or Broda in papers referred above.
10. I would advise you to read and consider to mention in your introduction a paper by Iakovlev et al. this paper is widely cited and describes by people involved in MECSEL research. It describes a design quite similar to the one you investigate. Compare your introduction and the one by Iakovlev. Iakovlev do not describe all the possible lasers architectures as you do but provides only the information which makes the reader to understand the improvements proposed by author of the paper Iakovlev, V., et al. "Double-diamond high-contrast-gratings vertical external cavity surface emitting laser." Journal of Physics D: Applied Physics 47.6 (2014): 065104. https://iopscience.iop.org/article/10.1088/0022-3727/47/6/065104
Errors to be corrected:
For a more spatially uniform laser microfabrication, a flat-top beam gives better the best results and more best control [2,3]. For fabrication high-aspect-ratio structures, a Bessel beam is required [4].
I would delete the words: “more” because I precedented sentences you say nothing about microfabrication its uniformity and its control, so what does it mean here “more” ? The word more serves to compare, for instance, you have something uniform then you do something special and it became more uniform.
L26 One is to shape the beam freely propagating behind the laser, to suit one’s needs [5], and another is to shape the pump beam to impose a different gain distribution and gain guiding in the resonator. This sentence is wrong because 1) the ward behind is unsuitable here (please check it meaning in a dictionary https://www.merriam-webster.com/dictionary/behind) consider beam freely propagating outside the laser resonator,
2) the lasers are not necessary optically pumped. Here you write about the pump beam for the first time. In precedent sentences you should state that you will concentrate on optically pumped lasers.
3) Shaping the pump beam to impose a different gain distribution is true but
and gain guiding in the resonator -is true only when the gain medium fill-up whole resonator what is not true, for instance, in disc lasers, also the pump beam can change the index of the gain medium thus change the guiding effect (but not gain guiding)
Consider: One is to shape the beam freely propagating outside the laser resonator, to suit one’s needs [5], and another is to shape the pump beam to impose a different gain distribution and/or guiding in the resonator.
In my previous review of your paper I advised you to concentrate in the introduction on optically pumped disc laser this would help you to be precise.
L28 Technically, this approach facilitates the light interaction with the gain media without mechanical or electrical manipulation of the resonator. The word light is necessary here. Reader has to know what interacts.
L45 The sentence: Their ability to emit light from the surface instead of from the edge (in more common semiconductor lasers) allows them to be clustered into large 2D arrays. Is not true, VCSELs can be fabricated as 2D array but VECSEL do not. VCSELs are electrically pumped VECSELs optically !
L47 A vertical surface-emitting laser comprises a thin amplifying medium made from quantum wells with Bragg mirrors formed on both sides to form a resonator [23,24]. This is not true, please read the references you provide! The VECSEL has only one DBR ! The MECSEL do not have the DBR at all !
L50 The short resonator design leads to gain guiding and index guiding effects [25,26]. This is not true ! VECSEL are type of disc laser with thin (usually um) gain medium and large (usually many cm long) external cavity. The light is controlled by external concave mirror, not gain or index guiding in a semiconductor heterostructure.
L51 In these lasers, transverse modes form due to the pump profile (this is true) and not from the geometrical structure of the cavity and the output coupling mirror (this is not true !) . In VECSEL and MECSEL the modes are determined by external cavity, that is by output coupling mirror position !
L 53 Vertical external-cavity surface-emitting lasers (VECSELs) and their newer versions, the membrane external cavity surface emitting lasers (MECSELs) with the cavity extending from the monolithic semiconductor chip (containing the back mirror and active media), have the ability to both have higher beam quality and higher output powers [28] when compared to monolithic designs.
To make this sentence true change to
Vertical external-cavity surface-emitting lasers (VECSELs) and their newer versions, the membrane external cavity surface emitting lasers (MECSELs) have the ability to both have higher beam quality and higher output powers [28] when compared to VCSEL monolithic designs.
L64 The PhCSEL is not VECSEL with photonic crystal. VECSEL is optically pumped PhCSEL is electrically pumped. VECSEL without PhC achieve higher power and better beam quality then PhCSEL !
Compare the ref 26 and papers by Susumu Noda https://www.nature.com/articles/nphoton.2014.75 this paper has 390 citation.
L66 ref 34. By Lukowski et al. is not about photonic crystal on top of VECSEL, The optical element in Lukowski VECSEL resonator change the beam mode but not improves the brightness
the ref 35 is also unsuitable here try the paper by Susumu Noda https://www.nature.com/articles/nphoton.2014.75
L70 In this work, we investigate whether a novel mode of operation can be achieved via structured optical pumping. A diode-pumping approach and a surface-emitting membrane semiconductor active medium characterize our particular approach. Those two sentences should be corrected. You may try
In this work, we demonstrate that a novel brightness of a disc laser mode of operation can be achieved improved via structured optical pumping. For this purpose we usie modified diode-pumping beam approach and a surface-emitting membrane semiconductor active medium characterize our particular approach.
L91 a word laser or amplifier is missing after “broad area semiconductor” broad area semiconductor laser (or amplifier) makes sense. Broad area semiconductor does not makes sense, sound as description of a wafer.
L114 what was the mode/fiber dimeter of your laser pump ?
There are an errors in the caption of Figure 1. There are no PP, BS in the figure a
An active area is encased in head spreaders should be heat
L122 should be propagation of the laser beam Laser itself do not propagates !
L159 there is no verb in the sentence The acquired conversion efficiency of η = 25 %.
L165 should be The oscillating resonator fundamental mode must be… not The oscillating
fundamental resonator mode must be…
L170 please correct “slight decrease in energy power in the central “
L172 please correct “from with an increase in the pump power is shown in (Supplementary video 1).
L282/283 Therefore, the resonant mode size decreased and was only coupled to the fundamental mode. Correct this sentence resonant mode cannot couple to fundamental mode
L287 correct …resonating beam’s energy power onto…
Comments on the Quality of English Languageneeds to be improved
Round 3
Reviewer 2 Report
Comments and Suggestions for Authors
Dear Authors
please consider same minor improvments of your manuscript.
There is
L33 However, they can be used to pump lasers as in the case of the application of pump laser diodes [11,12]. The achievement in spatial quality and brightness of the optically pumped laser trumps the power losses that would occur if one would simply spatially filter the laser diode beam.
Mayby would be better
However, they can be used to pump lasers as in as, for instance, in diode pumped solid state laser set-ups. The achievement improvments in spatial quality and brightness of the optically pumped laser trumps the power losses that would occur if one would simply spatially filter the laser diode beam.
L38 There is an alternative case However there are same cases where a Gaussian shape is not the best option for all
applications, and other types of transverse intensity patterns are desirable. For a more uniform laser ablation process as compared to a Gaussian pattern, a flat-top beam gives better results and more control as compared to a Gaussian pattern [13,14].
L45 Another, A a more exotic choice sophisticated (advances) option, is to shape the pump beam to impose a different gain distribution and/or guiding in the resonator.
L50 Extracavity modulation is more related suitable for better…
The sentence seems to be to long. You may consider to change:
Intracavity is mostly used for mode selection where gain or loss can be spatially controlled to select the desired mode [21–24] of the laser output, and higher energy extraction efficiency from the lasing medium [25,26], by using top-hat or flat-top beam profiles when the beam propagates through the gain region.
to
Intracavity is mostly used for mode selection since the gain or loss can be spatially controlled to select the desired mode [21–24] of the laser output. It also permits for higher energy extraction efficiency from the lasing medium [25,26], by using top-hat or flat-top beam profiles of the beam propagating through the gain region.
L55 The A selection of passive [27] or active [28,29] elements can be used for this purpose.
Before the sentence L63 The compact vertical cavity surface-emitting lasers (VCSELs)…. an introduction should be written which explains to readers why do you start to write about VCSEL/VECSEL etc. For instance you can write:
The flexibility of surface emitting semiconductor laser architecture gives a unique opportunity to test aforementioned beam shaping/brightness improvment techniques. The compact….
L68 In these lasers, transverse modes form due to the pump current spread out profile and i.e., how the charges are guided through the cavity [39]. The large aperture VCSEL as well as 2D arrays although are capable of high power emission provide multimode beams thus low brightness. However, optically pumped vertical external cavity surface-emitting laser…
in the sentence all should be plural:
L72 These versions are typically pumped with a laser diodes featuring its their specific transverse intensity profiles.
L75 These structures should have better power scalability due to spatially uniform optical pumping, much smaller thermal stresses [43] and better thermal management [44].
L78 Photonic crystal (PhC) spatial filters are a potential solution to suppress multimode operation in microcavity semiconductor surface emitting lasers, increasing the output beam spatial quality and its brightness [45–47].
L80 The other alternative is This happens also when photonic crystal structures is integrated with the gain medium as in PCSELs (photonic crystal surface emitting laser)[48,49].
L81 PCSELs are well-known for their capacity capability to generate…
L83 which is a major improvement compared to traditional conventional semiconductor surface-emitting lasers.
However, due to the complex photonic crystal structures and demanding fabrication techniques, they have difficulty sustaining it is difficult to sustain this high-power output across a wide range of applications.
L88 but they have traditionally had there were issues with thermal management and producing high power output emission in a compact form, which have been gradually addressed with technological advances.
Until now you have explained how the different designs of semiconductor surface emitting laser deal with the beam brightness issue so before the sentence which provides new idea tested in medium a introduction is needed
Lastly, the most exotic periodic structures can be created by interfering with multiple beams at an angle. Colloidal quantum
You may try with
Lastly, the emission brightness can be improved by modification of pump beam. The periodic gain structures can be created by interfering multiple pump beams aim at the different angles. Colloidal quantum...
Comments on the Quality of English Language
proofreading is necessery
Round 4
Reviewer 2 Report
Comments and Suggestions for Authors
Dear Authors
sorry I bother you so many times but I think you have nice experimental results and after same improvements of your manuscript you will have good paper in high impact factor journal, hopefully highly cited.
I think your paper is ready to be published. Although professional proof reading would be good sine I am not native English speaker I am not an expert. You can try https://www.proof-reading-service.com/en/ this is commercial, it is not very chip, but can be recommended.
Last problem I see in the introduction is that in your description you jump from extracavity to intracavity and vice versa.
L45 - 50 Intracavity…
L50 ….. (intracavity modulation). Extracavity…..
L51 Intracavity…
In this paper you can leave as it is (or change is up to you). For readers especially those less familiar with the subject it is always easier to understand to follow your arguments when you do not jump. Your goal is every reader understand your achievements.
There are two sentences which provide the same (similar) information which are separated in text by other sentences
Intracavity is mostly used for mode selection where gain or loss can be spatially controlled to select the desired mode [21 –24] of the laser output, and higher energy extraction efficiency from the lasing medium [ 25, 26], by using top-hat or flat-top beam profiles when the beam propagates through the gain region. A selection of passive [27] or active [28 ,29 ] elements can be used for this purpose. Intracavity modulation is beneficial because the resonator acts as a filter and purifies the oscillating mode [30 –33 ].
I think that resonator acts as a filter and purifies the oscillating mode = is mostly used for mode selection ………. to select the desired mode [21 –24]
Am I correct ?
Good lack with your research !
Comments on the Quality of English Language
proofreading would help
